**Review**

# Toward an understanding of glucose metabolism in radial glial biology and brain development

Madeline G Andrews[1] , Caroline A Pearson[2]

**Decades of research have sought to determine the intrinsic and extrinsic mechanisms underpinning the regulation of neural progenitor maintenance and differentiation. A series of precise temporal transitions within progenitor cell populations generates all the appropriate neural cell types while maintaining a pool of self-renewing progenitors throughout embryogenesis. Recent technological advances have enabled us to gain new insights at the single-cell level, revealing an interplay between metabolic state and developmental progression that impacts the timing of proliferation and neurogenesis. This can have long-term consequences for the developing brain's neuronal specification, maturation state, and organization. Furthermore, these studies have highlighted the need to reassess the instructive role of glucose metabolism in determining progenitor cell division, differentiation, and fate. This review focuses on glucose metabolism (glycolysis) in cortical progenitor cells and the emerging focus on glycolysis during neurogenic transitions. Furthermore, we discuss how the field can learn from other biological systems to improve our understanding of the spatial and temporal changes in glycolysis in progenitors and evaluate functional neurological outcomes.**

## Introduction

### Progenitor populations in the developing cortex

During the early phases of neocortical development, proliferative neuroepithelial cells (NECs) undergo amplifying symmetric divisions to contribute to the tangential expansion of the neocortex (Fig 1) (Taverna et al, 2014; Casas Gimeno & Paridaen, 2022). At embryonic day (E)11.5 in mice, NECs undergo an epithelial to mesenchymal-like transition into radial glia (RG), so called because of their distinctive bipolar morphology with apical and basal processes spanning the radial extent of the neocortex. RG reside within the ventricular zone (VZ) with their apical processes in direct contact with the ventricle and cerebrospinal fluid (Fig 1A). Earlier-born RG predominantly divide symmetrically to self-renew and expand the population and asymmetrically divide to give rise to early-born cortical neurons. Nascent neurons migrate along the RG basal process and form the cortical plate (Fig 1A) (Casas Gimeno & Paridaen, 2022). At E13.5 in mice, RG begin to undergo more asymmetric divisions to generate basal progenitors, including intermediate progenitors (IPs) and outer radial glia (oRG, also termed basal (b) RGs), as well as excitatory glutamatergic neurons (Fig 1A) (Dimou & Götz, 2014; Taverna et al, 2014; Beattie & Hippenmeyer, 2017). This transition is associated with a lengthening of the cell cycle; short cell cycles favor self-renewal, whereas longer cell cycles favor neurogenic divisions (Takahashi et al, 1995; Cai et al, 1997; Beattie & Hippenmeyer, 2017). This switch to generating basal progenitors forms a second progenitor niche, the subventricular zone (SVZ), sandwiched between the VZ and cortical plate (Fig 1A). The human cerebral cortex has a prolonged period of RG expansion, and the size of the SVZ and the diversity of basal progenitors it contains are expanded (Fig 1B). Across mammalian neocortical development, RG are the building blocks of the mature cortex. RG are not only responsible for its radial expansion, as they serve as the glial scaffold upon which differentiating cells migrate. In addition, RG generate all cortical excitatory neurons, astrocytes, and oligodendrocytes (Noctor et al, 2001; Rakic, 2009).

The transition to neurogenic RG divisions, the production of BPs, and the resulting radial expansion of the cortex create two spatially distinct progenitor microenvironments: the VZ and the SVZ (Fig 1A). Within each zone, there is varying metabolite availability based on the proximity to cerebrospinal fluid in the ventricle and the perfusion of the cortex by the developing vasculature (Fig 1A). The microenvironment in which neural progenitor cells are born and reside plays a critical role in their maintenance and differentiation. The progenitor zones comprise several elements: growth factors, cell–cell contacts, extracellular matrix factors, blood vessels, and cerebrospinal fluid. These factors play critical roles in the regulation of many facets of RG biology, including proliferation, differentiation, morphology, polarity, and intercellular organization (Long & Huttner, 2019; Fame & Lehtinen, 2020; Ferent et al, 2020). RG

[1]School of Biological and Health Systems Engineering, Arizona State University, Tempe, AZ, USA   [2]Center for Neurogenetics, Brain and Mind Research Institute, Weill Cornell Medicine, New York, NY, USA

Correspondence: mgandrew@asu.edu; cap4010@med.cornell.edu

labeled across mouse neurogenesis (E12–E15) collected for single-cell transcriptomics led to the observation that intrinsic programs regulate early RG, and later in development, RG transition to environment-sensing "extrinsic" programs (Telley et al, 2019). This increased sensing of the extracellular environment coincides with shifts in the structure of the cortex, active angiogenesis, and changes in oxygen and glucose availability. As we will discuss, blood vessels deliver oxygen and a range of metabolites, including glucose, to meet the metabolic demands of proliferating and differentiating RG.

## The development of cortical vasculature creates a highly dynamic radial glia microenvironment

In the mouse, from E8.5 to E10, and in the human from gestational week (GW) 6–7, angioblasts form a blood vessel network (termed the perineural vascular plexus) that envelops the developing neural tube (Paredes et al, 2018). From E11.5 onwards, endothelial cells migrate dorsally from the ventral telencephalon, perfusing the mouse cortex in a ventral-to-dorsal fashion (Fig 1A) (Vasudevan et al, 2008; Rosenstein et al, 2010; Tata & Ruhrberg, 2018; Karakatsani et al, 2019; Vogenstahl et al, 2022; "Cross-talk between blood vessels and neural progenitors," no date). Early in embryogenesis, symmetrically dividing RG produce lactate and express proteins from the Wnt family of developmental signaling molecules to encourage vasculogenesis/angiogenesis (Haigh et al, 2003; Daneman et al, 2009; Dong et al, 2022). Between E12.5 and E13.5, there is a significant elaboration of the vasculature as vessels coalesce and perfuse the expanding cortex (Vasudevan et al, 2008; Javaherian & Kriegstein, 2009; Lange et al, 2016; Komabayashi-Suzuki et al, 2019). Vegf signaling is a master regulator of angiogenesis in the cortex (Haigh et al, 2003; Raab et al, 2004). From E12.5 onwards, RG themselves express VegfA, thereby directing the vascularization of the mouse cortical progenitor microenvironment (Rosenstein et al, 2010; Mackenzie & Ruhrberg, 2012).

## Paracrine signaling by endothelial cells influences neural progenitor differentiation

Cortical angiogenesis forms a honeycomb network of vessels in the SVZ, whereas the VZ is relatively avascular (Fig 1A). IPs and oligodendrocyte precursors develop close associations with the vasculature in the SVZ (Javaherian & Kriegstein, 2009; Komabayashi-Suzuki et al, 2019). Within the VZ, sprouting endothelial tip cells preferentially form contacts between filopodia and mitotic RG, and the extracellular matrix factor, integrin $\beta$8, appears to regulate this connection to promote RG self-renewal (Komabayashi-Suzuki et al, 2019). Ventral telencephalic RG share this close connection with tip cell filopodia. However, in this setting, high filopodia density and Vegfa signaling are required to prolong the cell cycle of apically bound RG, favoring neuronal differentiation (Di Marco et al, 2020). This variance may be because of temporal differences in angiogenesis across regions, the more restricted number of RG and IPC in the ventral telencephalon, and the eventual fate of cell types in each region. The ventral telencephalon is a smaller, transient structure that produces the migratory inhibitory interneurons that populate the dorsal cortex. The metabolic requirements of these

discrete progenitor and neuronal subtypes may also contribute to the observed regional differences.

Co-culture experiments demonstrate that mouse endothelial cells promote mouse RG self-renewal and inhibit differentiation by activating the Notch signaling pathway. In contrast, smooth muscle cells do not influence RG differentiation (Shen et al, 2004). Further co-culture studies have demonstrated that soluble factors from embryonic mouse endothelial cells promote RG proliferation, whereas direct contact with endothelial cells promotes RG differentiation (Gama Sosa et al, 2007). A recent study defined the unique transcriptomic signatures of vascular cell, endothelial, and mural subtypes during human prenatal angiogenesis (Crouch et al, 2022). This study elegantly demonstrated that discrete signaling mechanisms, including collagen, laminin, and midkine, impact intercellular interactions (Crouch et al, 2022). As in mice, human endothelial tip cells are present in the VZ during neurogenesis, and in vitro co-culture of tip cells within human forebrain organoids promotes neurogenesis (Crouch et al, 2022). In these studies, the authors functionally assess the metabolic activity of primary human endothelial cells and demonstrate increased metabolic activity between GW16 and GW24 (Crouch et al, 2022). Thus, the specific role of endothelial cell signaling during RG differentiation is highly dependent upon several factors, including vascular and endothelial cell subtypes, developmental stage, and spatial proximity. Further exploration of the temporal relationship between vascular cells and RG across brain regions and between species will provide new insights into the contribution of endothelial cells to the progression of RG self-renewal versus differentiation.

## Metabolites fueling ATP production in radial glia

The elaboration of the cortical vasculature increases the availability of metabolites in the microenvironment that RG can use to produce ATP. Alternative metabolites, including amino and fatty acids, can feed into the tricarboxylic acid (TCA) cycle and contribute to anabolic programs promoting RG proliferation (reviewed by Namba et al [2021]). For instance, glutaminolysis converts glutamine into alpha-ketoglutarate, which feeds into the TCA cycle. In mouse and human RG, this process promotes RG proliferation during early neurogenesis and BP expansion at mid-neurogenic stages in humans (Journiac et al, 2020; Namba et al, 2020). These studies and others suggest that evolutionary changes in metabolism may fuel the expansion of neural progenitor proliferation, contributing to the growth and size of the human brain (Florio et al, 2015; Pontzer et al, 2016). Although a comprehensive view of the metabolic activities that regulate neurogenic progression is essential, this review will focus on the role of oxygen availability and glucose metabolism in RG proliferation and cell fate decisions.

Glucose is transported into and out of cells by membrane-bound glucose transporters through facilitated diffusion (Fig 2). RG express Glucose Transporter 1 (Glut1), whereas neurons express Glut3 (Choeiri et al, 2002; Lange et al, 2016; Dong et al, 2022). Several enzymatic steps break down one 6-carbon glucose molecule into two 3-carbon pyruvate molecules. Intermediate products include fructose 1,6 bisphosphate, glyceraldehyde 3 phosphate, and phosphoenolpyruvate, converted by glucokinase, phosphofructokinase, and pyruvate kinase, respectively (Lunt & Vander Heiden,

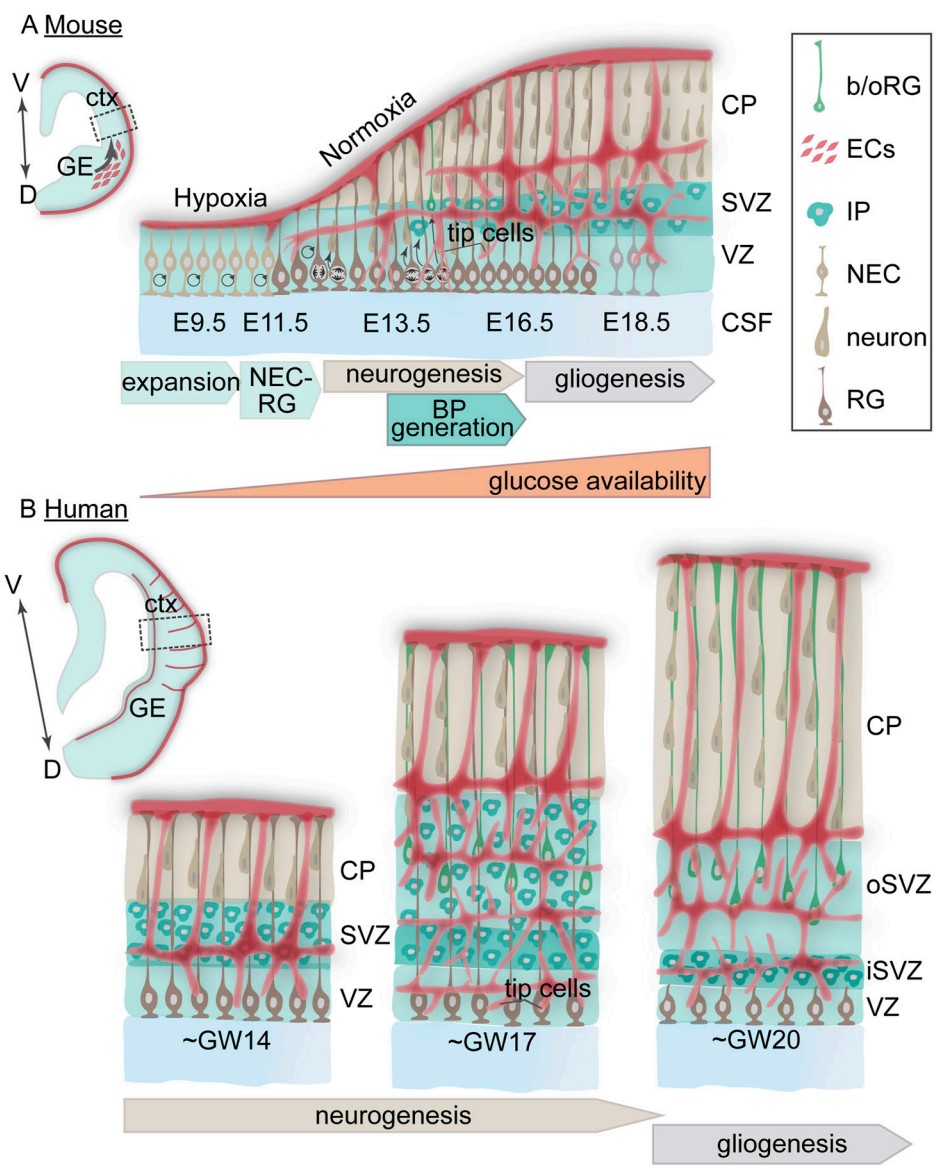

Figure 1. Overview of cortical neurogenesis and angiogenesis during mouse and human corticogenesis.

**(A)** Early self-renewing neuroepithelial cells (NEC) proliferate in an avascular environment and contribute to the tangential expansion of the cortex. At embryonic day (E) 11.5, NECs transition into radial glia (RG), and endothelial cells from the ventral telencephalon migrate into the cortex in a ventral to dorsal fashion. From E12.5 onwards, the cortex becomes increasingly vascularized through angiogenic programs orchestrated partly by signals from RG. Dividing RG at the apical surface form close interactions with endothelial tip cell filopodia. The increased vascularization of the cortex coincides with the transition of RG from predominantly self-renewing symmetric divisions to asymmetric neurogenic divisions, generating neurons and basal progenitors, including intermediate progenitors (IPs) and outer RG in the subventricular zone (SVZ). From E16.5 onwards, RG transition to gliogenic programs, and RG basal processes form close associations with blood vessels. **(B)** At gestational week (GW) 14, the human fetal cortex consists of three discrete progenitor zones. The ventricular zone (VZ) consists of ventricular radial glia interacting with endothelial tip cells. The SVZ comprises intermediate progenitors, and neurons populate the cortical plate (CP). By GW16, the SVZ has two distinct zones: the inner SVZ (iSVZ) and outer SVZ (oSVZ) composed of IPs and outer radial glia (oRG). This region is significantly expanded in the human cortex. The onset of gliogenesis occurs from GW20 onwards. The iSVZ population of IP persists into these later stages of embryogenesis. b/oRG, basal/outer radial glia; CSF, cerebrospinal fluid; ctx, cortex; ECs, endothelial cells; GE, ganglionic eminence; IP, intermediate progenitor; iSVZ, inner subventricular zone; NEC, neuroepithelial cell; oSVZ, outer subventricular zone; RG, radial glia; SVZ, subventricular zone; VZ, ventricular zone.

2011; Kierans & Taylor, 2021). Without oxygen or in low-oxygen conditions, lactate dehydrogenase A (LDHA) converts pyruvate to lactate, generating four ATP molecules per glucose molecule (net gain is two ATP molecules) (Fig 2). Lactate is transported out of the cell by monocarboxylate transporters (MCTs) (Kierans & Taylor, 2021).

Under normoxic or aerobic conditions, pyruvate dehydrogenase converts pyruvate into acetyl–CoA that enters the TCA cycle and generates NADH for oxidative phosphorylation (OxPhos) (Fig 2). The TCA cycle breaks down acetyl–CoA through a series of reactions and intermediates, including citrate, succinate, fumarate, malate, and oxaloacetate (Lunt & Vander Heiden, 2011). This process produces a net gain of 36 ATP molecules per glucose (Fig 2). Lactate entering the cell can be converted into pyruvate and enter the TCA cycle, augmenting ATP generation from glucose breakdown. OxPhos takes place in the mitochondria and links the TCA cycle to the generation

of ATP through a series of oxidative steps. Although mitochondrial-dependent respiration generates increased amounts of ATP molecules, it is a slower process and produces potentially damaging reactive oxygen species (ROS) (Lunt & Vander Heiden, 2011).

## Hypoxia and anaerobic glycolysis promote early RG self-renewal and expansion

Seminal research has begun to determine how the transition from anaerobic states to normoxia/aerobic glycolysis impacts RG transitions in vivo. Given their hypoxic microenvironment, early RG (~E11–E12.5) metabolize glucose through anaerobic glycolysis to generate ATP (Fig 3A; (Khacho et al, 2016; Lange et al, 2016; Komabayashi-Suzuki et al, 2019; Telley et al, 2019; Dong et al, 2022; Sakai et al, 2022). Anabolic pathways require glycolysis to produce energy to build tissues and perform cellular activities like cell

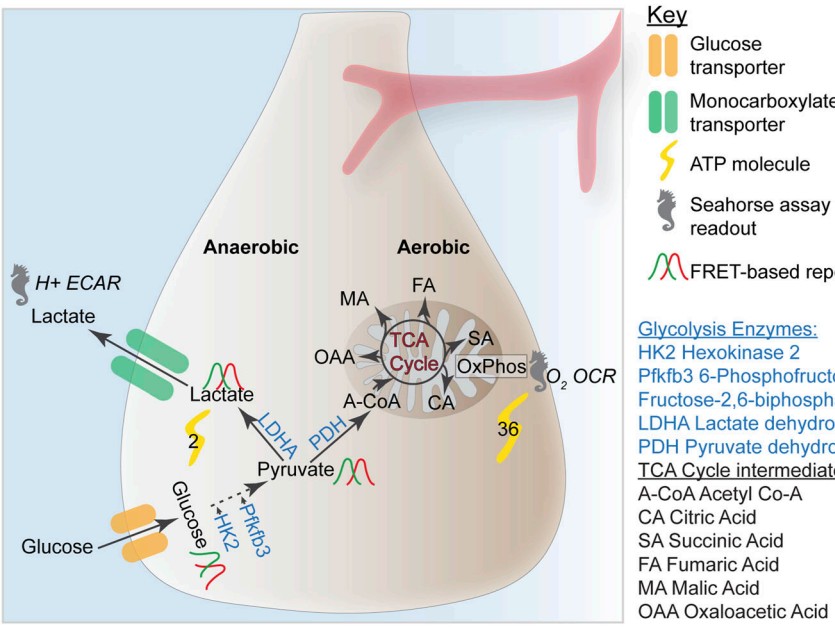

**Figure 2. Glucose metabolism during anaerobic and aerobic glycolyses.**
Membrane-associated glucose transporters transport environmental glucose into the cell. Through a series of enzymatic reactions, including hexokinase 2 (HK2) and 6-phosphofructo-2-kinase/fructose-2, 6-biphosphatase (Pfkfb3), glucose is converted into pyruvate. In anaerobic conditions, pyruvate is converted into lactate by lactate dehydrogenase A (Ldha) and transported out of the cell by monocarboxylate transporters. In aerobic conditions, pyruvate is transported into the mitochondria and converted into Acetyl CoA, which enters the tricarboxylic acid cycle. Oxidative phosphorylation (OxPhos) uses NADH produced by the tricarboxylic acid cycle via the electron transport chain. Anaerobic glycolysis produces a net gain of two ATP molecules, whereas aerobic glycolysis generates a net gain of 36. Seahorse metabolic flux assays measure oxygen consumption rate and extracellular acidification rate as readouts of OxPhos and lactate production, respectively. Förster resonance energy transfer-based reporters measure intracellular glucose, pyruvate, and lactate levels.

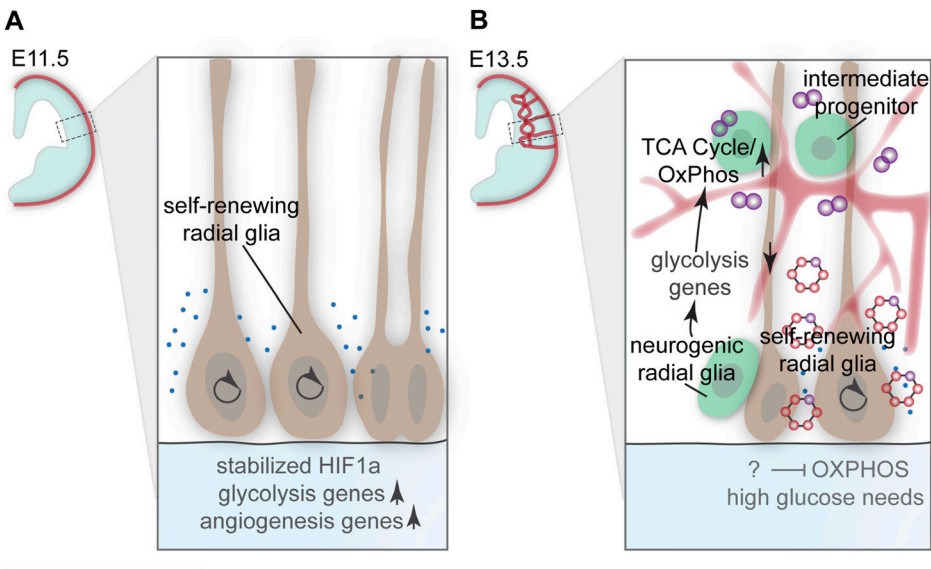

**Figure 3. Glycolytic programs used by cortical progenitors at early and mid-neurogenic stages of development.**
**(A)** In the absence of vasculature, at E11.5 in mice, radial glia use anaerobic glycolysis. HIF1a target genes are up-regulated, including genes involved in glycolysis and angiogenesis. Lactate produced during anaerobic glycolysis is transported out of the cell, encouraging angiogenesis. Hypoxic conditions promote self-renewing symmetric RG divisions. **(B)** At E13.5 in a vascularized environment, RG divide asymmetrically to generate neurogenic radial glia that become intermediate progenitors and neurons. Intermediate progenitors depend on oxygen, generate energy through the tricarboxylic acid cycle/OxPhos, and exhibit decreased expression of glycolysis genes compared with self-renewing radial glia. Radial glial differentiation requires this transition to aerobic glycolysis. Self-renewing radial glia rely on anaerobic glycolysis and are acutely dependent on glucose. The mechanisms self-renewing radial glia use to repress OxPhos and withstand hypoxia are unknown.

division. Although the net production of ATP is lower in anaerobic glycolysis than in aerobic glycolysis, it is a more rapid method of energy production (Lunt & Vander Heiden, 2011). In the mouse cortex, perturbing cortical angiogenesis prolonged early hypoxia, which increased RG self-renewal and decreased neurogenic divisions that generate IPs and neurons. Concomitantly, hypoxia-inducible factor 1 alpha (HIF1α) target gene expression increased, including glycolysis genes (Lange et al, 2016). Similarly, exposing pregnant dams to decreased oxygen, between E10.5 and E12.5, increased mitotic RG, although, in this model, there was increased vascularization as a result of hypoxia (Komabayashi-Suzuki et al, 2019). These studies suggest that the early hypoxic

environment in the mouse cortex and anaerobic glycolysis maintain the self-renewal capacity of RG in the VZ (Fig 3A; genetic manipulations are summarized in Table 1).

An in-depth understanding of the metabolic state of neural cell types in human development in utero is challenging to assess. However, in vitro approaches provide tractable access to cell-specific molecular programs, like adherent culture of induced pluripotent stem cell (iPSC)-derived neural progenitor cells or 3D neural organoids. During differentiation, human iPSC-derived neurons express lower levels of many, but not all, glycolysis genes than RG. In vitro loss of glycolysis genes, *Hexokinase 2* (*HK2*) and *LDHA*, in progenitors results in decreased proliferation (Zheng et al, 2016). Constitutive co-activation of HK2 and LDHA increases the proportion of progenitors transitioning to aerobic respiration and concurrently increases neuronal death (Zheng et al, 2016).

### Neurogenic RG transition to aerobic glycolysis and OxPhos

At mid-neurogenic stages (E13.5–E15.5), the VZ comprises asymmetrically dividing RG that employ different metabolic mechanisms to self-renew or differentiate (Fig 3B). This review will refer to these as self-renewing RG and neurogenic RG, respectively. As described above, a hallmark of self-renewing RG is the reliance on anaerobic glycolysis. However, neurogenic RG are less driven by rapid anabolic cell division as they migrate into the SVZ and differentiate into IPs and neurons (Fig 3B). In the mouse, at E13.5–14.5, gene expression analyses indicate neurogenic RG and IPs rely on mitochondrial ATP generation, that is, TCA cycle and OxPhos. Self-renewing RG express higher levels of glycolysis genes than asymmetrically dividing neurogenic RG (Fig 3B) (Khacho et al, 2016; Lange et al, 2016; Dong et al, 2022). In the mouse brain, targeted knockdown of the glycolysis gene *Pfkfb3* results in increased neuron generation at the expense of RG and transcriptional decreases in genes associated with glycolysis. However, the authors did not report corresponding increases in TCA Cycle/OxPhos-related genes or increased oxygen consumption rate (OCR) (Kalucka et al, 2015), and *Pfkfb3* knockdown does not specifically target anaerobic glycolysis (Lange et al, 2016). Thus, although anaerobic glycolysis appears to regulate RG self-renewing divisions, whether the onset of neurogenesis requires the cessation of anaerobic glycolysis requires additional investigation.

**Table 1. Summary of phenotypes observed by genetic manipulations impacting glycolysis.**

| Mutation/perturbation | Phenotype | Citations | Species | System |
|---|---|---|---|---|
| *Gp124$^{-/-}$* | Early hypoxia, decreased angiogenesis, increased anaerobic glycolysis, increased RG self-renewal | Lange et al (2016) | mouse | In vivo |
| *HIF1α$^{flox/+}$; Emx1$^{Cre/+}$* | Increased neurogenesis, increased IPs, decreased RG | Lange et al (2016) | mouse | In vivo |
| *HIF1α $^{flox/flox}$; Sox2$^{Cre/+}$* | Neuronal cell death, hypoplasia | Sakai et al (2022) | mouse | In vivo |
| *HIF1α $^{flox/flox}$; Nestin$^{Cre/+}$* | Neuronal cell death, hydrocephalus | Tomita et al (2003) | mouse | In vivo |
| *HIF1α$^{fl/fl}$; Nestin$^{Cre\ ER}$* | No impact on RG viability. Neurons are susceptible to hypoxia | Candelario et al (2013) | mouse | In vitro |
| *Mfn1/2$^{-/-}$* | Severe mitochondrial fragmentation, increased neurogenic divisions | Khacho et al (2016) | mouse | In vivo |
| *Drp1$^{-/-}$* | Excessive mitochondrial elongation, increased self-renewing divisions | Khacho et al (2017) | mouse | In vivo |
| *HK2/LDHA* knockdown (shRNA) | Decreased NPC proliferation | Zheng et al (2016) | human | In vitro |
| Hyper-oxygenation (mid-neurogenesis) | Increased IPs, increased neurogenesis, no effect on RG | Wagenführ et al (2015) | mouse | In vivo |
| Exposure to hypoxia (mid-neurogenesis) | Decreased IPs, premature differentiation, no effect on vRG or oRG | Pașca et al (2019) | human | In vitro |
| Moderate hyperglycemia | Increased differentiation, early RG cell cycle exit | Ji et al (2019) | mouse | In vivo |
| Severe hyperglycemia | Decreased differentiation. Increased VZ | Rash et al (2018) | mouse | In vivo |
| TIGAR knockdown | Increased glycolysis ending in lactate, increased proliferation, decreased differentiation, decreased proneural gene expression | Zhou et al (2019) | mouse | In vivo |
| *Ldha–/–* | Decreased early RG proliferation, increased neuronal differentiation | Dong et al (2022) | mouse | In vivo/in vitro |
| Lactate injection | Increased early RG proliferation, decreased differentiation | Dong et al (2022) | mouse | In vivo/in vitro |
| Culture with lactate | Increased RG (glucose induces differentiation) increased IPs | Álvarez et al (2016) | mouse | In vivo/in vitro |
| Leigh syndrome (*PDH$^{-/-}$; MT-ATP6/PDH$^{-/-}$; DLD$^{-/-}$*) | Decrease in neuroepithelial formation. Abnormal mitochondrial morphology Increased astrogliogenesis | Romero-Morales et al (2022) | human | In vitro |

In vitro metabolomic assays have been used to link transcriptional changes with changes in functional readouts of glycolysis, including lactate production and OCR (Fig 2). These studies have demonstrated that as RG differentiate, lactate concentration in the media decreases, indicating decreased reliance on anaerobic glycolysis (Lange et al, 2016; Álvarez et al, 2016; Dong et al, 2022). Studies have demonstrated that in the mouse, early NECs (E8.5) predominantly respire anaerobically, and OCR increases as they age and transition into RG (E10.5–E13.5) (Fame et al, 2019; Dong et al, 2022). Functional assays can further dissect the glycolytic differences between self-renewing and neurogenic RG throughout neurogenesis and whether these changes are correlative or causal with the progression of neurogenic programs (Fig 2).

These studies indicate that and an asymmetry in RG fate decisions at mid-neurogenic stages, there is an asymmetry regarding glycolysis (Fig 3B). The model these studies propose suggests that self-renewing RG rely on anaerobic glycolysis, whereas neurogenic RG preferentially generate ATP via the TCA cycle/OxPhos. However, it is unclear how RG achieve this asymmetry. And for driving anabolic processes required for cell division, many studies have demonstrated that metabolic programs play a role in directing cell cycle progression (reviewed by Kalucka et al [2015]). Cell cycle length in RG influences cell fate choices; longer cell cycles favor asymmetric neurogenic divisions, whereas shorter cell cycle promotes symmetric self-renewing divisions (Takahashi et al, 1995; Cai et al, 1997; Beattie & Hippenmeyer, 2017). One potential mechanism for further exploration is how differing glycolytic programs influence RG cell cycle progression.

## Oxygen levels differentially impact progenitor populations at mid-neurogenic stages

Another potential mechanism is the further subdivision of the VZ into apical hypoxic and basal normoxic zones. In 2019, Komabayashi-Suzuki et al elegantly described the temporal and spatial progression of the cortical vasculature (Komabayashi-Suzuki et al, 2019). This study reported that apical RG lining the VZ are in an avascular environment and expressed HIFα. In contrast, the basal VZ is vascularized, and basal RG do not express HIF1α (Komabayashi-Suzuki et al, 2019). This description differs from previous reports that the entire VZ is avascular and hypoxic, and RG do not express HIF1α (Javaherian & Kriegstein, 2009; Lange et al, 2016). These differences may be because of the sensitivity of the techniques or reagents used to assess vasculature patterns or protein expression.

Genetic manipulations in mouse embryos shed some light on the contribution of oxygen levels and HIF1α signaling in RG at mid-neurogenic stages (Table 1). Conditional removal of both HIF1α alleles in mouse RG severely impacts cortical angiogenesis, resulting in hypoplasia, microcephaly, and neuronal cell death (Tomita et al, 2003; Lange et al, 2016; Sakai et al, 2022). Removing one HIF1α allele is sufficient to rescue the angiogenesis phenotype and allow investigation of the role of HIF1α signaling in RG (Lange et al, 2016; Komabayashi-Suzuki et al, 2019). Conditional loss of one HIF1α allele in RG at mid-neurogenic stages promotes IP proliferation but not at the expense of RG, suggesting that mid-neurogenic RG are unaffected by the reduction in HIF1α (Komabayashi-Suzuki et al, 2019). In contrast, conditional removal of one HIF1α allele in early RG reduces the expression of HIF1α target genes, including

glycolysis genes, and increases neurogenesis at the expense of self-renewal (Lange et al, 2016).

Similarly, hyperoxic conditions selectively increase BP populations in the vascularized SVZ. Increased maternal oxygen promotes the proliferation of IPs and oRG populations in mouse and gyrencephalic ferret embryonic brains. Yet, the RG population remains unaffected by increasing oxygen levels (Wagenführ et al, 2015). These studies suggest that mid-neurogenic RG reside in an avascular/hypoxic environment and are resistant to hypoxia. In further support of this, in vitro culture shows that at mid-neurogenesis, RG are more resistant to hypoxia than neurons, and conditional removal of HIF1α does not impact viability, suggesting that HIF1α signaling does not underlie the mechanism through which RG can withstand prolonged hypoxia (Candelario et al, 2013). These studies do not support the model that the VZ is subdivided into hypoxic and normoxic zones, suggesting that differential modes of glycolysis and asymmetric cell fate decisions are not linked to spatial variances in the oxygen level. In addition, it is still unknown how RG can withstand prolonged hypoxia throughout neurogenesis.

However, these studies clearly show that oxygen levels impact basal progenitor populations. In models of human brain development, changes in oxygen levels also influence basal progenitor development. Forebrain organoids, derived from human pluripotent stem cells, were cultured under hypoxic conditions (1% oxygen), resulting in a significant reduction in the number of IPs without impacting ventricular radial glia (vRG) or oRG numbers (Paşca et al, 2019). A more in-depth study is required to determine how oRG cells generate ATP and whether they are more metabolically similar to vRG or IP as they develop in a presumptively more oxygen-rich SVZ microenvironment later in cortical development. Hypoxia gene expression and post-mitotic markers are increased in IP in hypoxic conditions, suggesting diminished oxygen levels cause IPs to exit the cell cycle and differentiate into neurons before they do in control conditions. Thus, whereas hypoxic conditions do not impact human RG, the balance between IP proliferation and differentiation is sensitive to oxygen levels during human development (Wagenführ et al, 2015; Paşca et al, 2019).

An additional consideration in human corticogenesis is the presence of two discrete domains where IPs reside, termed the inner (i) SVZ or the outer (o) SVZ (Fig 1B). The apically localized subset of IPs within the iSVZ persists for a longer duration, into late neurogenesis, whereas the oSVZ is more transient (Pebworth et al, 2021). The vascular perfusion of the SVZ coincides with the expansion of the oSVZ domain and the growth of the IP population (Pebworth et al, 2021; Crouch et al, 2022). Human studies demonstrate that IP production is sensitive to oxygen levels (Paşca et al, 2019); however, the mechanism by which the iSVZ IP pool perdures in this relatively avascular region is unknown. The prolonged presence of the iSVZ could reflect spatially distinct vasculature patterns or a subset of IP that, similarly to RG, are less sensitive to oxygen levels.

## The balance between RG self-renewal and differentiation is sensitive to changes in glucose concentration

In addition to oxygen dynamics, the development of the cortical vasculature also increases the availability of environmental glucose

(Fig 3B). The intracellular glucose level in RG during neurogenesis is unclear; however, mid-neurogenic RG appear to be acutely dependent on extracellular glucose (Rash et al, 2018). Pregnant dams fed a high-fat diet model severe hyperglycemia because of maternal obesity (Rash et al, 2018). In this model, the offspring have a smaller and thinner cortex, and both RG and IP populations expand at the expense of neurons (Rash et al, 2018). In contrast, an increase in neurons is seen in the offspring of dams with moderate hyperglycemia (Ji et al, 2019). In the obesity model, maternal blood glucose levels were twice as high as those reported by Ji et al (2019), suggesting that glucose concentration plays a pivotal role in regulating RG response.

Moderate maternal hyperglycemia models demonstrate that high glucose levels cause the accumulation of hydrogen peroxide, increase oxidative stress, and promote the expression of histone acetylation proteins (Yu et al, 2016; Bai et al, 2018; Ji et al, 2019; Li et al, 2019). Greater histone acetylation promotes the transcription of proneural genes, resulting in early cell cycle exit and premature differentiation. Overexpression of superoxide dismutase to counteract the effects of oxidative stress or blocking histone deacetylase P300 can rescue the effects of moderate hyperglycemia in RG (Yu et al, 2016; Bai et al, 2018; Li et al, 2019). The glycolytic activity under these conditions is undetermined; therefore, it is unclear whether epigenetic changes and the promotion of neurogenesis coincide with increased reliance on the TCA cycle/OxPhos. Future studies should define the temporal and causal relationships between glucose concentration, metabolic activity, histone acetylation, and neurodevelopmental progression.

During normal mouse development, TP53 inducible glycolysis and apoptosis regulator (TIGAR) is up-regulated in the cortex, including in RG, from E12.5 onwards (Zhou et al, 2019). TIGAR inhibits anaerobic glycolysis, promotes the conversion of lactate to pyruvate, and increases levels of Acetyl CoA. In addition to being an essential substrate in the TCA cycle, Acetyl CoA regulates histone acetylation. Knockdown of *TIGAR* in NSCs in vitro reduced acetylation at the promoters of proneural genes, resulting in decreased neuronal differentiation (Zhou et al, 2019). The mechanism underpinning TIGAR up-regulation in the cortex is unknown. However, *TIGAR* is a known transcriptional target of the HIF1α target gene, *p53* (Chen et al, 2003). TIGAR is ubiquitously expressed throughout the cortex at mid-neurogenic stages, so it remains to be determined whether self-renewing RG repress TIGAR-mediated OxPhos.

Although changes in oxygen, glucose availability, and chromatin accessibility affect RG self-renewal and brain development (summarized in Table 1), it is not clear how glycolysis regulates cell cycle dynamics, asymmetric cell divisions, and subsequent cell fate decisions. As we will discuss, improving our understanding of RG glucose metabolism through cutting-edge metabolomic technology established in other systems will be an essential step forward.

### Experimental approaches to study glucose metabolism in the regulation of cellular behavior

Transcriptional changes do not always translate into corresponding protein decreases, nor are they accurate readouts of metabolic choices (Walls et al, 2020). Metabolic dynamics of stem cells have been studied extensively in the context of cancer. We can leverage in vitro and in vivo metabolic assays developed in the cancer field to evaluate how metabolism regulates neural development and contributes to dysfunction in neurodevelopmental disorders.

Limited real-time extracellular flux assays can unpack the rate of glycolysis and metabolic switching from anaerobic glycolysis to the use of aerobic glycolysis during neurogenesis. Discrete populations of cell types in the mouse and human brains assayed using Seahorse assays have demonstrated changes in extracellular acidification rate (anaerobic glycolysis) and OCR (OxPhos) across different developmental stages (Fig 2) (Fame et al, 2019; Komabayashi-Suzuki et al, 2019; Crouch et al, 2022; Dong et al, 2022). Extracellular flux assays can be applied across cultures of isolated cell types or ex vivo tissue explants to determine the dynamics of secretion of limited numbers of metabolites from living cells. This assay is particularly useful for comparing large-scale changes across time, real-time changes in response to perturbations or control versus mutant conditions. However, the results are cumulative and are less useful for disentangling cell type-specific interactions. Additional approaches are required to clarify glucose transport dynamics and how glucose anabolism/catabolism affects individual cells and the broader cortical niche.

Mass spectrometry (MS) is routinely used to quantify and compare glycolysis and TCA cycle substrates in vivo and in vitro (Fig 2) and can be used in combination with $^{13}C_6$-labeled glucose to determine the metabolic fate of glucose. A recent study used this technique with explanted mouse embryonic cortices to trace labeled glucose in E10.5 and E13.5 RG (Dong et al, 2022). Nuclear magnetic resonance spectrometry has been used to detect secreted and intracellular metabolites and can be used within intact tissues when paired with magnetic resonance imaging (Yan & Xu, 2018; Emwas et al, 2019). In rats, nuclear magnetic resonance has been used to model prenatal exposure to chemicals associated with the development of autism in vivo. This technique successfully identified several metabolic biomarkers related to prenatal insult within distinct brain regions (Abreu et al, 2021). However, sensitivity, metabolite selection, and sample preparation all inform which metabolomic approach is most appropriate for sample processing and acquisition.

Spatial metabolomics is essential to retain metabolite information in intact tissues. As we have described, the tissue-specific niche is vital for the appropriate development of relevant cell types, and those cell–cell relationships inform both metabolic and developmental states. Matrix-assisted laser desorption/ionization MS uses laser excitation of endogenous metabolites in the tissues and deconvolution of laser intensity to identify metabolite presence in a given topographical location (Fujimura & Miura, 2014). Matrix-assisted laser desorption/ionization-MS has successfully been applied to adult mouse brain, liver, and kidney metabolites (Sugiura et al, 2012; Kim et al, 2013), and more recently in early mouse embryos to assess metabolomic changes during neural tube closure (Miyazawa et al, 2017; Vaughn et al, 2021). Other approaches pair fluorescent in situ hybridization and mass spectrometry, for instance, to compare metabolic heterogeneity in the fetal kidney and kidney organoids and show that maturation-associated fatty acid metabolism is not recapitulated in vitro (Wang et al, 2022).

Microscopy approaches are lower throughput but have increased cellular resolution and can be paired with genetically

encoded metabolic reporters or dyes to provide cell or organelle-specific metabolic dynamics. Fluorescently labeled glucose analogs such as 2-NBDG and 6-NBDG have been used to compare glucose uptake and glycolytic activity across cell types in vitro and in vivo (Zou et al, 2005; Barros et al, 2018). 2/6-NBDG are transported into cells and undergo phosphorylation but do not degrade further. Thus, fluorescence accumulation is a readout of glucose uptake. However, canonical glucose transporters do not transport 2/6-NBDG, and uptake is slower than environmental glucose (Hamilton et al, 2021). Förster resonance energy transfer biosensors are a powerful tool for monitoring metabolite production and utilization using genetically encoded reporters. These reporters can be visualized in vivo or in vitro to evaluate the comparative abundance of glycolysis substrates, including glucose, lactate, and pyruvate. For instance, a förster resonance energy transfer-based pyruvate reporter termed *Pyronic* determined the quantity of pyruvate to measure transport dynamics within and between neural cell types in in vitro mouse cells and in vivo *Drosophila* larvae (San Martín et al, 2014; Arce-Molina et al, 2020). Microscopy approaches can orthogonally evaluate glycolysis metabolites to pinpoint the metabolic state of a particular cell type within the developing cortical progenitor niche. In combination, we can apply these approaches to developing neural cell cultures or tissue samples to define the metabolic programs used by dynamic populations of progenitors and validate model efficacy for interrogating features of neurodevelopmental disorders.

# Conclusions and Significance

To understand disease states, we need to achieve increased resolution of the metabolic activity of neural progenitor populations and how metabolic state guides neurodevelopment. Many metabolic disorders, including mutations in genes regulating glucose transport, mitochondrial disorders, maternal diabetes, fetal hyperglycemia, and obesity, are associated with increased prevalence of epilepsy, autism, and ADHD (Yazdy et al, 2010; Krakowiak et al, 2012; Linder et al, 2015; Márquez-Valadez et al, 2018; Aviel-Shekler et al, 2020). The consequences of changes to glucose levels and relevance to disease states require functional interrogation.

Over the last decade, advances in neural organoid culture and their use in human disease modeling have enabled researchers to model metabolic disorders and associated neurodevelopmental defects. Leigh syndrome is a metabolic disorder causing bilateral lesions in the brain and clinically manifests with seizures and movement abnormalities. This disorder is associated with gene mutations regulating the mitochondrial production of ATP, including components of the electron transport chain (Bakare et al, 2021). A recent study used patient iPSC-derived neural organoids to interrogate the developmental features of three patient lines with mutations in pyruvate dehydrogenase and dihydrolipoyl dehydrogenase (Romero-Morales et al, 2022). Phenotypically, patient lines have impaired developmental hallmarks, including changes in size, cortical disorganization, mitochondrial defects, and increased proportions of astrocytes. Future studies that interrogate functional changes in metabolism would inform the mechanisms driving Leigh syndrome and other metabolism-related neurodevelopmental disorders.

Despite recapitulation of many robust developmental programs in neural organoids, transcriptomic comparisons between organoid-derived cells and their endogenous counterparts identified alterations in metabolic gene expression associated with glucose metabolism, oxygen availability, and electron transport (Pollen et al, 2019; Bhaduri et al, 2020). However, in defined metabolic analyses of neural organoid datasets, metabolic changes driving cluster identity are restricted to a limited number of cell types (Uzquiano et al, 2022; Vértesy et al, 2022). Both apically localized vRG and an unspecified population of excitatory neurons were associated with increased expression of glucose metabolism and hypoxia genes, compared with a primary reference dataset. At the same time, the remaining cells within the culture are unaffected by the metabolic state (Uzquiano et al, 2022). Identified changes in metabolism are likely an artifact of culture conditions because of suboptimal medium composition and cellular location within the interior of the organoid, a region known to be hypoxic and lacking vasculature. Efforts toward neural organoid vascularization are ongoing (Cakir et al, 2019; Wang et al, 2021; Crouch et al, 2022; Sun et al, 2022). Further dissection of impaired metabolic programs and the impact on developmental state may provide insight into their role in normal development. In addition, although these studies have improved our understanding of the importance of metabolism in neurodevelopment, current interrogations of metabolic states in human in vitro cultures have been predominantly limited to transcriptional approaches, and functional assays will increase the resolution of these relationships.

The increased size of the human brain implies the evolutionary refinement of metabolic requirements needed to produce a more significant number and variety of neural cell types. In a comparative study between mouse and human neural development, the pharmacological activation of mitochondrial metabolism increased the maturation rate in human neurons in vitro and after mouse transplantation in vivo, whereas inhibition slowed maturation rates in mice. By increasing mitochondrial activity, neurons shifted toward greater use of OxPhos rather than anaerobic metabolism, as measured by functional readout of OCR, gene expression, and protein abundance changes. Neurons with increased mitochondrial activity also exhibited increased morphological complexity and activity measures associated with increased maturation (Iwata et al, 2023). Metabolic transition states appear to be essential for regulating RG division, neurogenic differentiation, expansion of basal progenitors, and ultimately, fate determination and maturation of neuronal populations. The dynamics of how RG metabolize glucose and oxygen and the differences in these processes between humans and our recent evolutionary ancestors can provide essential insights into the development and function of the central nervous system. Understanding how alterations lead to different developmental outputs can continue to shed light on how metabolism impacts brain health in neurodevelopmental disease states.

# Glossary

• Aerobic: a process that requires oxygen.

•Anaerobic: a process that does not require oxygen or oxygen is unavailable.

•Apical/ventricular radial glia: the neural stem cells of the developing cerebral cortex bound to the apical membrane and reside in the ventricular zone.

•Asymmetric division: differentiating division; when a progenitor divides along the oblique/horizontal plane, one daughter cell remains a progenitor, and one daughter cell differentiates.

•Basal progenitor: progenitor cells that reside basal to the ventricular zone. Basal progenitors include outer radial glia and intermediate progenitor cells.

•Cell fate: the terminal differentiated identity of a progenitor cell.

•Differentiation: the process by which a neural stem cell or multipotent progenitor progressively exits the cell cycle and adopts its terminal identity.

•Early neurogenesis: the early stage of neurogenesis in which RG typically divide symmetrically, between E10.5–E12.5 in the mouse and GW13–15 in humans.

•Extracellular acidification rate (ECAR): an indicator of lactate production, a consequence of anaerobic glycolysis.

•Glycolysis: the pathway by which glucose is broken down into pyruvate.

•HIF signaling: HIF1$\alpha$ enters the nucleus under hypoxic conditions and binds to hypoxia response elements (HRE). Activation of HIF signaling results in the transcription of genes that increase glycolysis, prevent oxidative stress, and promote angiogenesis.

•Hypoxic: insufficient or low quantities of oxygen available in the local environment.

•Intermediate progenitor: unipotent neurogenic progenitor cell that produces neurons.

•Late neurogenesis: end of neuronal differentiation before the onset of gliogenesis between E15.5–E16.5 in mice and GW19–20 in humans.

•Mid-neurogenesis: the stage in which RG typically divide asymmetrically, resulting in the peak generation of neurons between E12.5–E14.5 in mice GW16–18 in humans.

•Mural cells: vascular cells, including pericytes and smooth muscle cells that line the endothelium of developing vasculature.

•Neurogenic: promoting the production of neurons or refers to RG that will differentiate rather than self-renew.

•Normoxic: typical oxygen conditions.

•Outer radial glia: the basal radial glial population that resides in the outer subventricular zone of the developing cortex and is more prominent in gyrencephalic species, like primates, than in lissencephalic species, like rodents.

•Oxidative phosphorylation (OxPhos): aerobic metabolic pathway; the final step in cellular respiration where the electron transport chain produces ATP.

•Oxygen consumption rate (OCR): measurement to quantify the change in oxygen during glycolysis; indicative of OxPhos.

•Proliferation: the expansion of progenitor cell populations through cell division.

•Radial glia: the neural stem cells of the developing cerebral cortex that give rise to basal progenitors, excitatory neurons, astrocytes, and oligodendrocytes.

•Self-renewal: division in which a progenitor replaces itself. This can be achieved through symmetric or asymmetric division.

•Subventricular zone (SVZ): progenitor zone where basal progenitors divide and reside. This zone is expanded in the human neocortex.

•Symmetric division: a self-renewing division where a progenitor divides in half along the vertical plane to produce two progenitors.

•TCA cycle: the second stage of cellular respiration, which occurs after glycolysis and before oxidative phosphorylation. The TCA cycle yields NADH and FADH2, which OxPhos then uses to generate ATP molecules.

•Vegf signaling: vascular endothelial growth factor signaling, the regulatory signaling pathway of angiogenesis.

•Ventricular zone: the progenitor niche adjacent to the lateral ventricle where radial glia reside.

•Wnt signaling: developmental signaling pathway that regulates proliferation and patterning of neural progenitors.

# Acknowledgements

We thank M Elizabeth Ross and C Dyevich for scientific discussions and feedback. MG Andrews was supported by NIH Award R00MH125329, Brain and Behavior Research Foundation (BBRF) NARSAD Young Investigator Grant #29155, and Arizona Biomedical Research Centre New Investigator Award RFGA2022-010-16. CA Pearson was supported by R01NS105477, the Sackler Brain and Spine Research Institute at New York Presbyterian/Weill Cornell Medicine Research Grant, the American Epilepsy Society Junior Investigator Award, and the Glut1-Deficiency Foundation.

## Author Contributions

MG Andrews: conceptualization, resources, supervision, funding acquisition, visualization, and writing—original draft, review, and editing.

CA Pearson: conceptualization, resources, supervision, funding acquisition, visualization, project administration, and writing—original draft, review, and editing.

## Conflict of Interest Statement

The authors declare that they have no conflict of interest.

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
