## [Reviewer comments · Life Science Alliance]

Life Science Alliance

Toward an understanding of glucose metabolism in radial glial biology and brain development.

Madeline Andrews and Caroline Pearson

DOI: <https://doi.org/10.26508/lsa.202302193>

Corresponding author(s): Caroline Pearson, Weill Cornell Medicine

Review Timeline:

Submission Date:	2023-05-31
Editorial Decision:	2023-07-31
Revision Received:	2023-09-16
Editorial Decision:	2023-09-20
Revision Received:	2023-09-20
Accepted:	2023-09-26

Transaction Report:

July 31, 2023

Re: Life Science Alliance manuscript #LSA-2023-02193-T

Dr. Caroline Alayne Pearson
Weill Cornell Medicine
413 69th street east
Belfer Building
New York, NY 10021

Dear Dr. Pearson,

Thank you for submitting your manuscript entitled "Toward an understanding of glucose metabolism in radial glial biology and brain development" to Life Science Alliance. The manuscript was assessed by expert reviewers, whose comments are appended to this letter. We invite you to submit a revised manuscript addressing the Reviewer comments.

Thank you for this interesting contribution to Life Science Alliance. We are looking forward to receiving your revised manuscript.

Sincerely,

B. MANUSCRIPT ORGANIZATION AND FORMATTING:

Reviewer #1 (Comments to the Authors (Required)):

In this review manuscript, the authors summarize the recent advances in the metabolic regulation of neocortical development, with a specific focus on glycolysis. While the contents are informative, the authors need to revise the manuscript substantially because of the following reasons.

1. Since cell metabolism is a network, not independent pathways, focusing only on glycolysis and OxPhos will provide a biased view of the progenitor cell metabolism. For example, glycolysis is not the only one metabolic pathway that fuels the TCA cycle. Activation of other pathways, such as beta oxidation and glutaminolysis, can fuel the TCA cycle, thus potentially influence the activity of glycolysis. Therefore, the authors need to discuss other metabolic pathways more while keep focusing on glycolysis.
2. There are many sentences that are difficult to read. This problem is not only due to the grammatical issues, but also due to the insufficient description of terms. Examples are listed as follows: "Early RG express lactate....." (page 3). Does this mean the early RG produce lactate?; What is the "committed RG" (page 5)?; It is difficult for the reviewer to understand the exact meaning of "A more in-depth study is required to determine whether oRG, like vRG, generate ATP differently to IPs, despite being in an oxygenated microenvironment." (page 8); What does "prematurely differentiated" mean (page 8)?; All cells are relying on metabolism. Therefore, the reviewer thinks that "cancer cells rely heavily on metabolic processes" does not make any specific points (page 11); Please elaborate on the following sentence "By increasing mitochondrial activity, neurons shifted toward greater use of OxPhos" (page 15).

Minor comments

1. It is better to specify the species used in the studies mentioned every time.
2. There are several mistakes in citations.
3. The authors need to be careful about citing non-peer-reviewed articles, because these manuscripts have not been evaluated properly.

Reviewer #2 (Comments to the Authors (Required)):

This is a review manuscript on glucose metabolism in radial glial cells in the developing brain. This is an important and timely topic and there is a great interest in the community in how metabolism influences development. However, the review is written rather like listing of different publications instead of a more integrative mode in which to summarize which findings agree and integrate these into a conceptual view and report which data may be non-convincing and/or outliers. A second major point to consider is that it focuses entirely on neurogenesis during development, but it maybe of great interest to the reader to compare glucose metabolism of radial glial cells in development and as adult neural stem cells, where there are interesting differences.

More specific suggestions:

- p.3, 1st paragraph: "Early RG express lactate" - it is impossible to express lactate, the authors may refer to "have high levels of lactate"
- p.3, 2nd paragraph: This is an example where just manuscripts and their findings are listed without conceptualizing. Emphasize similar findings and highlight where results differ, e.g. between brain regions and explain why this could be the case - what is different in RGs and Ips between these regions... see e.g. Pilz et al. 2013
- p.4, 2nd paragraph: there are no citations on huge claims of vascular regionalization and their molecular signatures and also the relevance of these statements in the context of RG glucose metabolism isn't made clear
- p.4-6: glycolysis is needed for anabolic pathways and hence for cell division. This is not mentioned but should be incorporated especially for the changes observed between early, fast proliferating, and later, slower proliferating RGCs (e.g. on page 6, 2nd paragraph)
- p.5, bottom: when mitochondrial dynamics are reduced, ROS increases - what is cause and consequence in the findings reported on the bottom of the page
- p.6, middle of the 2nd paragraph: the sentence about the lactate catabolism has an intrinsic contradiction referring to the BEPCK-M pathway - please double-check is its correct

p.8, 3rd paragraph: statement that high glucose levels decrease histone acetylation proteins is wrong - it should be "increase" as also highlighted on p. 7

p.9, 3rd paragraph: it is likewise wrong that maternal obesity would model hypoglycemia - it models hyperglycemia.
The last part of the review on methods to analyze metabolism is written very well and provides a good overview.

We would like to thank the reviewers for their helpful and insightful comments. We have endeavored to address each point and correct mistakes where appropriate. We believe the reviews have helped us improve our literature review's content and clarity. Below, we have outlined our changes to address each reviewer's comments.

Reviewer #1 (Comments to the Authors (Required)):

In this review manuscript, the authors summarize the recent advances in the metabolic regulation of neocortical development, with a specific focus on glycolysis. While the contents are informative, the authors need to revise the manuscript substantially because of the following reasons.

We thank the reviewer for their feedback and have worked to considerably edit the manuscript to improve the scope of content and clarity of presentation.

1. Since cell metabolism is a network, not independent pathways, focusing only on glycolysis and OxPhos will provide a biased view of the progenitor cell metabolism. For example, glycolysis is not the only one metabolic pathway that fuels the TCA cycle. Activation of other pathways, such as beta oxidation and glutaminolysis, can fuel the TCA cycle, thus potentially influence the activity of glycolysis. Therefore, the authors need to discuss other metabolic pathways more while keep focusing on glycolysis.

We appreciate the reviewer's feedback and helpful suggestions. While we have retained our primary focus on glucose metabolism due to available research findings in the neurodevelopmental field, we have expanded our discussion to include more details about other metabolic pathways and relevant metabolites contributing to radial glial development. We have included a reference to a review in which the authors discuss alternative metabolic processes that contribute to neural progenitor metabolism. We have included a glossary of terms to clarify this discussion.

2. There are many sentences that are difficult to read. This problem is not only due to the grammatical issues, but also due to the insufficient description of terms. Examples are listed as follows: "Early RG express lactate....." (page 3). Does this mean the early RG produce lactate?; What is the "committed RG" (page 5)?; It is difficult for the reviewer to understand the exact meaning of "A more in-depth study is required to determine whether oRG, like vRG, generate ATP differently to IPs, despite being in an oxygenated microenvironment." (page 8); What does "prematurely differentiated" mean (page 8)?; All cells are relying on metabolism. Therefore, the reviewer thinks that "cancer cells rely heavily on metabolic processes" does not make any specific points (page 11); Please elaborate on the following sentence "By increasing mitochondrial activity, neurons shifted toward greater use of OxPhos" (page 15).

We apologize for the confusing terminology and insufficient description of these developmental processes. We have worked to better articulate and define developmental terms throughout the text, explicitly focusing on the abovementioned text lines. We have added a glossary to clarify the description of relevant neurodevelopmental and metabolic terms.

Minor comments

1. It is better to specify the species used in the studies mentioned every time.

We have added information about the relevant species used in described studies throughout the manuscript.

2. There are several mistakes in citations.

We apologize for this oversight and have updated our citation list.

3. The authors need to be careful about citing non-peer-reviewed articles, because these manuscripts have not been evaluated properly.

We have removed pre-prints from our citation list.

Reviewer #2 (Comments to the Authors (Required)):

This is a review manuscript on glucose metabolism in radial glial cells in the developing brain. This is an important and timely topic and there is a great interest in the community in how metabolism influences development. However, the review is written rather like listing of different publications instead of a more integrative mode in which to summarize which findings agree and integrate these into a conceptual view and report which data may be non-convincing and/or outliers. A second major point to consider is that it focuses entirely on neurogenesis during development, but it maybe of great interest to the reader to compare glucose metabolism of radial glial cells in development and as adult neural stem cells, where there are interesting differences.

We appreciate the reviewer taking the time to read our review on glucose metabolism in brain development and providing thoughtful feedback. We have extensively rewritten this review to better organize and conceptualize the scientific takeaways from the collective studies rather than listing the results. However, given the focus on the review to define the contribution of glucose metabolism to brain formation, tools for studying development, and contribution to metabolic neurodevelopmental disorders, we believe the discussion of adult neural stem cells is outside this review's scope.

More specific suggestions:

p.3, 1st paragraph: "Early RG express lactate" - it is impossible to express lactate, the authors may refer to "have high levels of lactate"

We apologize for this nomenclature mistake, as we were primarily referencing gene expression datasets. We have updated this phrase to describe lactate production.

p.3, 2nd paragraph: This is an example where just manuscripts and their findings are listed without conceptualizing. Emphasize similar findings and highlight where results differ, e.g. between brain regions and explain why this could be the case - what is different in RGs and Ips between these regions... see e.g. Pilz et al. 2013

We have greatly expanded our discussion of this and other findings. Specifically, we have included a comparison between dorsal and ventral telencephalon and the differences in cell development across neural regions.

p.4, 2nd paragraph: there are no citations on huge claims of vascular regionalization and their molecular signatures and also the relevance of these statements in the context of RG glucose metabolism isn't made clear

The discussion of human angiogenesis was previously focused on Crouch et al., 2022, as this is the first study on prenatal primary human vascular cell types, which we have additionally cited throughout the paragraph. We have modified this section considerably to expand the discussion of angiogenesis and its relationship to glucose availability and metabolism.

p.4-6: glycolysis is needed for anabolic pathways and hence for cell division. This is not mentioned but should be incorporated especially for the changes observed between early, fast proliferating, and later, slower proliferating RGCs (e.g. on page 6, 2nd paragraph)

We thank the reviewer for this suggestion and have clarified the relationship between anabolism and cell division throughout our RG glycolysis discussion and the relationship between these processes throughout the text.

p.5, bottom: when mitochondrial dynamics are reduced, ROS increases - what is cause and consequence in the findings reported on the bottom of the page

We have modified our discussion to focus less on mitochondrial/ROS dynamics and more on glycolysis. This statement has been removed due to limited relevance.

p.6, middle of the 2nd paragraph: the sentence about the lactate catabolism has an intrinsic contradiction referring to the BEPCK-M pathway - please double-check is its correct

After reflection, we apologize for this error and decided this content was superfluous. We have removed these details to avoid confusion and more clearly focus our discussion.

p.8, 3rd paragraph: statement that high glucose levels decrease histone acetylation proteins is wrong - it should be "increase" as also highlighted on p. 7

We apologize for this error and have updated the text to clarify this result.

p.9, 3rd paragraph: it is likewise wrong that maternal obesity would model hypoglycemia - it models hyperglycemia.

We apologize for this typo and have corrected it in the text.

The last part of the review on methods to analyze metabolism is written very well and provides a good overview.

We thank the reviewer for their feedback and appreciate their interest in this portion of the review. We have rewritten much of the other sections to streamline content and highlight major conclusions, limitations of cumulative study findings and outstanding questions.

September 20, 2023

RE: Life Science Alliance Manuscript #LSA-2023-02193-TR

Dr. Caroline Alayne Pearson
Weill Cornell Medicine
Brain and Mind Research Institute
413 69th street east
Belfer Building
New York, NY 10021

Dear Dr. Pearson,

Thank you for submitting your revised manuscript entitled "Toward an understanding of glucose metabolism in radial glial biology and brain development.". We would be happy to publish your paper in Life Science Alliance pending final revisions necessary to meet our formatting guidelines.

- please add an Author Contributions section to your main manuscript text
- please add callouts for Figures 1A and 3A-B to your main manuscript text

A. FINAL FILES:

B. MANUSCRIPT ORGANIZATION AND FORMATTING:

****Reviews, decision letters, and point-by-point responses associated with peer-review at Life Science Alliance will be published online, alongside the manuscript. If you do want to opt out of having the reviewer reports and your point-by-point responses**

displayed, please let us know immediately.**

Sincerely,

September 26, 2023

RE: Life Science Alliance Manuscript #LSA-2023-02193-TRR

Dr. Caroline Alayne Pearson
Weill Cornell Medicine
Brain and Mind Research Institute
413 69th street east
Belfer Building
New York, NY 10021

Dear Dr. Pearson,

Thank you for submitting your Review entitled "Toward an understanding of glucose metabolism in radial glial biology and brain development". It is a pleasure to let you know that your manuscript is now accepted for publication in Life Science Alliance. Congratulations on this interesting work.

Again, congratulations on a very nice paper. I hope you found the review process to be constructive and are pleased with how the manuscript was handled editorially. We look forward to future exciting submissions from your lab.

Sincerely,
